# Can network science reveal structure in a complex healthcare system? A network analysis using data from emergency surgical services

Katharina Kohler ,[1,2] Ari Ercole [1,2]

¹University Division of Anaesthesia, University of Cambridge, Cambridge, UK
²NHS Department of Anaesthesia, Cambridge University Hospitals NHS Foundation Trust, Cambridge, UK

**Correspondence to**
Katharina Kohler;
kk371@cam.ac.uk

## ABSTRACT

**Introduction** Hospitals are complex systems and optimising their function is critical to the provision of high quality, cost effective healthcare. Metrics of performance have to date focused on the performance of individual elements rather than the whole system. Manipulation of individual elements of a complex system without an integrative understanding of its function is undesirable and may lead to counterintuitive outcomes and a holistic metric of hospital function might help design more efficient services.

**Objectives** We aimed to use network analysis to characterise the structure of the system of perioperative care for emergency surgical admissions in our tertiary care hospital.

**Design** We constructed a weighted directional network representation of the emergency surgical services using patient location data from electronic health records.

**Setting** A single-centre tertiary care hospital in the UK.

**Participants** We selected data from the retrospective electronic health record data of all unplanned admissions with a surgical intervention during their stay during a 3.5-year period, which resulted in a set of 16 500 individual admissions.

**Methods** We then constructed and analysed the structure of this network using established methods from network science such as degree distribution, betweenness centrality and small-world characteristics.

**Results** The analysis showed the service to be a complex system with scale-free, small-world network properties. We also identified such potential hubs and bottlenecks in the system.

**Conclusions** Our holistic, system-wide description of a hospital service may provide tools to inform service improvement initiatives and gives us insights into the architecture of a complex system of care. The implications for the structure and resilience of the service is that while being robust in general, the system may be vulnerable to outages at specific key nodes.

## INTRODUCTION

Emergency surgical admissions make a significant contribution to pressures on hospitals and health services in general. This constitute an unpredictable high-resource patient group as their admissions are often

### Strengths and limitations of this study

► The analysis was based on a comprehensive electronic data set describing the emergency surgical services of a tertiary teaching hospital.
► The data were routinely collected, fully anonymised and non-identifiable; however, data errors and missing data were present.
► We used well-established network analysis methodology as appropriate for a complex system.
► To our knowledge, this is the first study aiming to characterise an inpatient service via network analysis.

characterised by intensive interventions and input from multiple teams. Such admissions are increasing: in October 2018, overall emergency admissions to hospital increased by 2% compared with the previous year.[1] Admission pressure can only be accommodated if inpatient flow and discharges are as efficient and timely as possible and this may have knock-on effects all the way back to accident and emergency (A&E) performance.[2] Increased emergency demand on a hospital affects elective services due to cancelled operations and increases in waiting times.[2 3]

Service improvement projects and metrics typically focus either on single-point interventions, specific periods within the patient journey or a designed subset of patients. Patient flow through hospitals has been investigated in various settings,[4–6] often in order to improve throughput in a specific setting such as the ED.[7 8] However, hospitals and the multiteam services within them are considered 'complex systems'.[9] Such highly interconnected and interdependent systems may respond counterintuitively to interventions targeting individual modules and a system-wide, holistic description is lacking.

Network science, where the components of a system are represented by nodes and the

connections between them by edges, provides a framework to investigate such systems and has been widely applied in other disciplines.[10–14] High-fidelity patient movement data are increasingly available in machine readable form making it available for modelling pathways in this way.

Recent papers used these methods in the medical services field to assess the structure of referral systems,[15] surgical care teams[9] and patterns of acute hospital admissions.[16] We wanted to assess whether we could apply network methodology to patient pathways throughout the hospital, specifically emergency surgical admissions and to gain insight into the characteristics and vulnerabilities in our service. We constructed a network representation of the emergency surgical service from patient movement data from our comprehensive Electronic Hospital Record (EHR). The overall aim was to obtain a fuller understanding of the real architecture of the system of emergency surgical admissions to allow us to identify important hubs and bottlenecks in the service structure.

## METHODS
### Data
We used fully anonymised routinely collected location data from the EHR data at Cambridge University Hospitals NHS Foundation Trust, a tertiary care hospital. This single centre serves as a district general hospital to its local population and as a referral centre for specific specialties such as major trauma, neurosurgery and paediatric surgery. All major surgical specialties except for cardiothoracic surgery are represented. We selected all unplanned admissions between January 2015 and July 2018 where the inpatient stay included a surgical procedure. This selection resulted in a data set of more than 16 500 individual admissions of both adult and paediatric patients. Among the adult patients, the most common admitting surgical specialties were trauma (23.5%), general surgery (13.7%) and neurosurgery with (9.3%). In addition to these surgical admitting specialties, a significant proportion (~21% of patients) were admitted under medical specialties but subsequently required a surgical procedure during their stay.

Patient movement data including dates and times for each inpatient journey were collated from admission until discharge. We also included location data from inpatient investigations such as CT scans and inpatient visits to other departments. We used locations and transfers to develop our network model as each transfer represents a use of resources and is thought to be based on a medically necessary decision—for example, a transfer to CT scan and back to the ward is related to the medical need for a scan and uses resources such as porters, ward staff and clinical staff to facilitate the event.

All data collected were extracted in a fully deidentified manner and stored securely. Under UK regulations, research ethics approval is not required for the reuse of anonymous routinely collected data for research. However, our project was approved by local institutional review.

### Patient and public involvement
No patients were involved in this research, but for any improvement activity that results from it PPI would be included.

### Model
#### Network terminology and variables
A comprehensive mathematical description of network science concepts has been given by Barabási and Posfai[10] and we include a short introduction to the concepts used in this paper. Briefly, a system can be represented by a network formed of nodes and edges. The nodes represent the components of the system and the edges represent the interactions or connections between the nodes. These edges can be undirected or directed, so that there are two different edges one from A to B and one from B to A. The number of connections a node has to other nodes is called the *degree* of the node. For directed networks, *in-degree* and *out-degree* can also be calculated by counting the connections in or out of a node. Additionally, the edge can be *weighted* by the strength of the connection to show the traffic between two nodes. The product of weight and degree is called the *strength* of an edge.

From these calculations, the distribution of degrees for all nodes can be constructed which contains information summarising the underlying fundamental structure of the network. If the network has only random connections, the degree distribution is a Poisson distribution; however, in most real-life networks, the degree distribution can be approximated by a log-normal or power-law distribution $p(k) \sim k^{-\gamma}$, where p is the probability distribution, k is the degree and γ is the exponent of the power-law.

Networks that exhibit a power-law distribution have a lot of nodes with a small degree and a long 'tail' of the distribution where a few nodes have disproportionately large degrees (hubs). These networks are called *scale-free*[17] and have a robust architecture as they are resistant to random node failure. This is because a randomly selected node most likely has a low degree, so that the impact of taking it out of the network is minimal. However, the presence of hubs makes scale-free networks vulnerable to targeted insults where a hub is compromised.[9 18]

It is then also possible to calculate a number of network properties including local and global *clustering coefficients* (the extent to which node neighbours are linked), *betweenness centrality* (identifying nodes with many shortest paths that are important to the overall functioning of the system), *flow hierarchy* (the extent to which flow is directed), *reciprocity* (the fraction of transfers that exist in both directions between two nodes) and *assortativity* (the tendency of nodes to be connected to nodes of similar degree in our case).

Furthermore, the combination of clustering coefficient, C, and shortest path length, L, can then be used to assess the network for *small-world* characteristics. Small-world

**Patient pathways:**

Three example patient pathways through the hospital

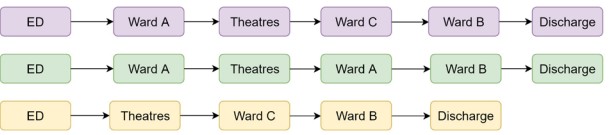

**Network based on the pathways above:**

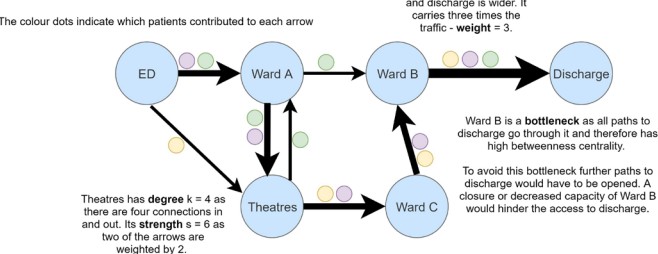

The colour dots indicate which patients contributed to each arrow

The edge between Ward B and discharge is wider. It carries three times the traffic - **weight** = 3.

Ward B is a **bottleneck** as all paths to discharge go through it and therefore has high betweenness centrality.

To avoid this bottleneck further paths to discharge would have to be opened. A closure or decreased capacity of Ward B would hinder the access to discharge.

Theatres has **degree** k = 4 as there are four connections in and out. Its **strength** s = 6 as two of the arrows are weighted by 2.

**Figure 1** Construction of the network. We show three illustrative patient pathways and how they are combined to construct the network representation using wards as nodes and transfers as edges. Several of the analysis parameters in the text are also explained, for example the degree of a node.

networks are a type of network with specific properties such as high degree of clustering and clique formation, short average path between the nodes and an abundance of hubs, nodes with many connections.[19–21] The measures $\sigma = \frac{\frac{C}{C_{random}}}{\frac{L}{L_{random}}}$ and $\omega = \frac{L_{random}}{L} - \frac{C}{C_{lattice}}$ measure how closely a network resembles the small world ideal by comparing it to random and lattice networks. The short path length and high clustering coefficient are thought to facilitate easy and efficient flow of information and team-work[9 22] within the system.

## Network construction

We used the timestamps to reconstruct each patient's journey which were in return used to construct the network (figure 1). Patient locations were assigned to nodes and the patient transfers to edges. The data extraction from the EHR, data cleaning and construction of the network model was performed using python, networkx,[23] igraph[24] and cytoscape.[25] Data acquisition and handling was approved locally both via the trust quality improvement department and the university data request system. No further ethical approval was needed as no patients were directly contacted.

We constructed both unweighted and weighted networks, where the weighting was the frequency of the transfer reflecting the traffic between two nodes. Since some specific wards could change their designation (eg, from one specialty to another) or be repurposed or closed without changing the underlying service, we subsequently created a 'categorised' network by replaced agglomerating physical locations into care categories (eg, 'acute medical ward').

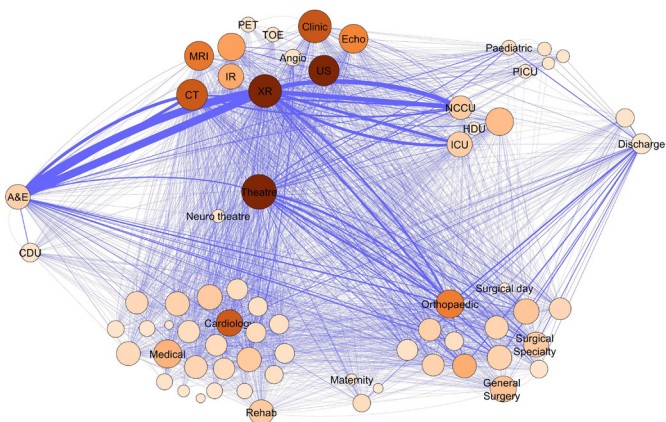

**Figure 2** Non-categorised network of emergency surgical admissions. The network of transfers shown grouped by clinical categories of care. The nodes are coloured by betweenness centrality—higher betweenness centrality is shown in deeper colour and sized by overall degree. The nodes were arranged to represent a patient journey from A&E admission (on the left) to discharge (on the right) and the nodes were grouped together to show the different types of locations. Some of the important nodes have been labelled to show the grouping with the medical wards at the bottom left, the surgical wards bottom right, investigations at the top, critical care areas and theatres in the middle and the paediatric services at the top right. Most nodes are left unlabelled for clarity with some indicative labels in each group. The ward abbreviations are explained in the supplementary table. A&E, accident and emergency.

## Analysis

We calculated the network characteristics such as degree distribution, strength distribution, flow hierarchy, global clustering and small-world properties described in the appendix. Similarly, we investigated the properties of individual nodes with the measures described above, weighted and unweighted degree, clustering coefficients and betweenness centrality.

## RESULTS
### Overall network and degree distribution

The total cleaned data set consisted of more than 16 500 individual admissions with 230 000 transfers. The 'non-categorised' network of all transfers of patients admitted via A&E is shown in figure 2. The wards and services are grouped together by type of area to better illustrate the variety of services involved in the care of these patients. Descriptions of the abbreviations used can be found in the online supplementary table. This overall network representation of our emergency surgical care network shows dense connectivity between the different wards and hospital services. Areas that are particularly densely connected (eg, theatres or radiology) most often have high overall network importance based on betweenness centrality. However, there are a few areas (eg, medical wards) which are well connected but less structurally important.

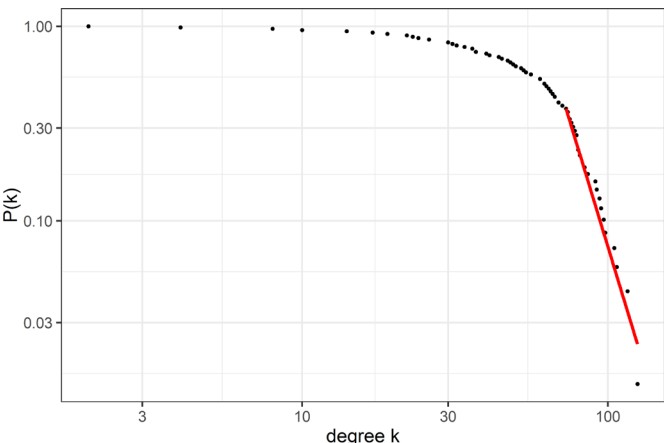

**Figure 3** Degree distribution. The degree distribution (as log–log plot) for our network of wards. The distribution shows a power-law behaviour at the right-hand tail of the distribution. The power-law fit (obtained with the package poweRlaw[28]) is shown in red with a γ of 6.1.

Figure 3 shows the degree distribution for our network. We used the methodology outlined in Clauset *et al*[26] to investigate the best fit to the tail of the distribution. The power-law fit generated using a bootstrap method[27 28] to determine the most representative fit to the tail of the distribution $P(k)=k^{-\gamma}$ is superimposed in red. We found $\gamma=6.18$ (95% CI: 6.14 to 6.26) with a p-value for the null hypothesis of 0.46, which means that a power-law is an appropriate fit.

The categorised network is shown in figure 4. Some of strongest connections are unsurprisingly between emergency department and radiology or CT scan—illustrating the need for initial investigations on a patient's admission to hospital. Other strong connections are more surprising. The presence of a connection from ED to the general medical wards and one from these wards to discharge shows that our patient cohort attended the medical wards both preoperatively and postoperatively. The betweenness centrality measure shows the importance of theatres,

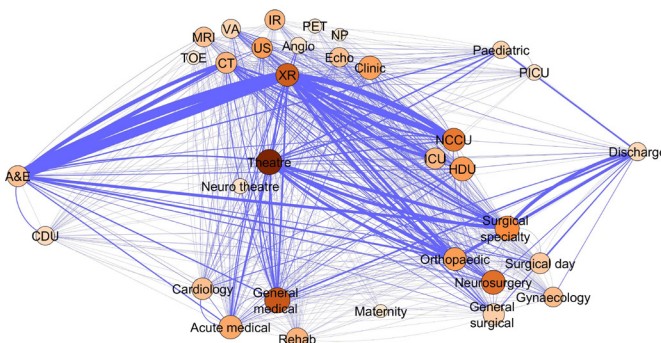

**Figure 4** Categorised network. The network of transfers shown grouped by area of care. Here, the nodes are categories rather than physical locations and as in figure 2 coloured by betweenness centrality—higher betweenness centrality is shown in deeper colour and sized by overall degree. The ward abbreviations are contained in the online supplementary table.

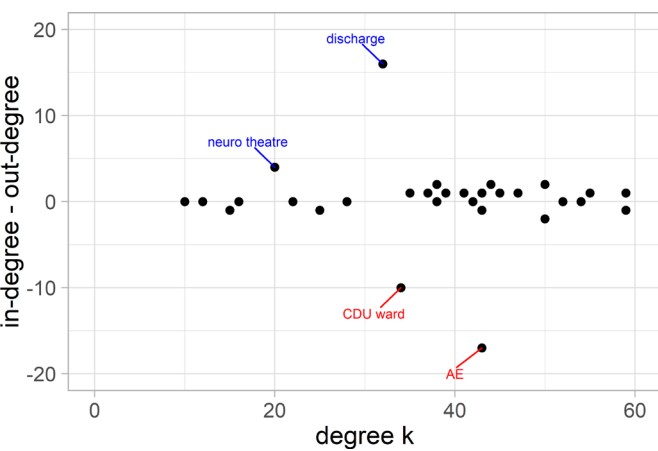

**Figure 5** Net connectivity of nodes. This shows the unweighted difference in in-degree and out-degree, net connectivity, versus overall degree k for our system of wards. The distribution wards such as A&E are in the lower half of the graph (red labels) and the receiving wards in the upper (blue labels). Most wards have a balanced traffic. The ward abbreviations are explained in the supplementary table. A&E, accident and emergency.

general medical and neurosurgical wards to the overall function of the system.

We found both our non-categorised and categorised networks had high reciprocity (with r~0.9 and r~0:8, respectively). This means that most paths exist in both directions and that transfers from one area to another are usually countered by transfers in the opposite direction—not necessarily the same patient. This balance reflects the operational need to fill all beds and spaces to allow for the high demand within the service.

### Degree analysis

The in-degree and out-degree balance is shown in figure 5 for the non-categorised network. Most wards fall on or near the zero-line showing that they receive from and send patients to a similar number of places; however, a few areas are so-called distributors (in red) or receivers (in blue). The clinical decision unit, a short stay medical ward that receives patients shortly after admission is a 'distributor' as it sends patients to a large variety of locations. The appearance of the neurosurgical theatres as a 'receiver' outlier was unexpected. It can be explained by the specialist nature of the neurosurgery service, which means that the patients tend to be transferred to a limited set of wards for postoperative care but due to bed-pressures, they may be cared for preoperatively in a range of settings.

### Weights

The relationship between the node strength s (weighted degree) and degree in figure 6. Part A of the figure shows the strength versus degrees k for the categorised network. The labelled nodes are selected either due to significantly higher traffic, represented by strength s, than expected by their connectivity (degree) or significantly lower.

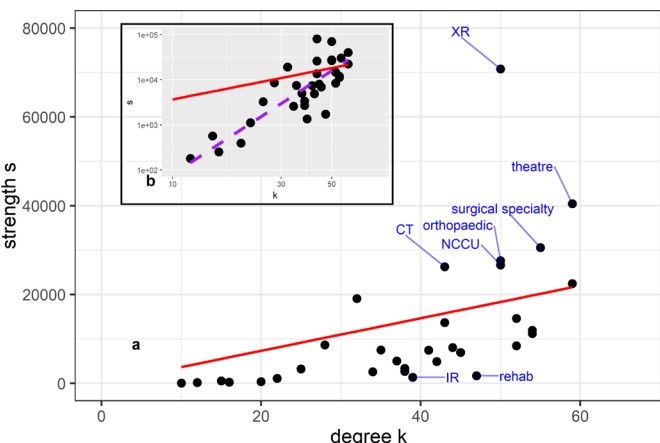

**Figure 6** Relation between traffic and connectivity. The figure explores the strength (weighted degree) versus degrees for the categorised system of wards. (A) The overall distribution of strength versus degree, where the labelled nodes are outliers with respect to their degree–strength relation, either significantly higher traffic than expected by their connectivity or significantly lower. The graph also shows in red the relation between strength and degree if the weights were uncorrelated. (B) The inlay shows the log–log distribution of strength versus degree with the uncorrelated distribution (red) and the power-law fit (purple dashed) showing that the strength grows faster than the degree. The ward abbreviations are explained in the online supplementary table.

In the absence of correlation between weight and degree, the strength of a node should be proportional to its degree[29] $s{\sim}k$ (shown by the red line in figure 6), but the data are not well fitted by this distribution, especially at lower degrees. The inlay figure B shows the log–log distribution of strength versus degree again with the uncorrelated line in red and additionally the power-law fit (purple dashed line) for $s{\sim}k^{\beta}$. The power-law is a much better fit with $\beta=2.1$ showing that the strength of the nodes grows significantly faster than the degree and that higher connectivity wards experience disproportionately more traffic.

## Assortativity

Assortativity $a$, which measures how similar the neighbours of a node are with respect to degree, falls between −1 and 1. We found our system to be dissortative with assortativity coefficients of $a=-0.20$ and $a=-0.12$ for non-categorised and categorised networks, respectively. The dissortative effect is also seen in figure 7 which shows the average degree of the nearest neighbours, $k_{nn}$, versus degree with the best linear fit overlaid in red. This means that on average, high-degree nodes are connected to nodes with lower degree and not on average to other highly connected nodes, thus exposing the network to a higher risk of disconnection should one of the highly connected nodes fail.[30]

## Betweenness centrality

Figure 8 shows a strong correlation of betweenness centrality with degree. The correlation is shown as a

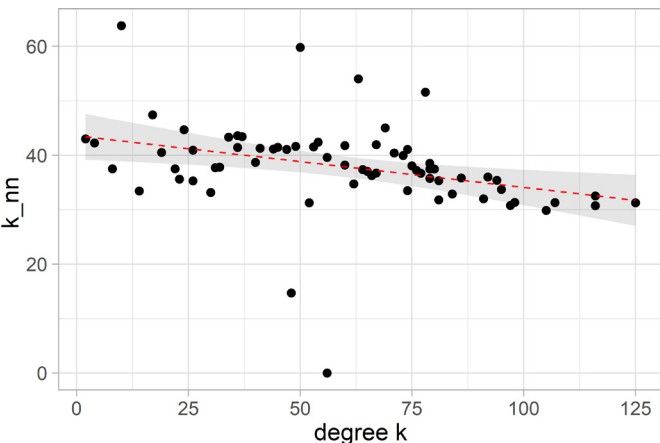

**Figure 7** Assortativity. The weighted nearest neighbour degree $k_{nn}$ versus degree k for the non-categorised network. It shows the dissociative behaviour of the network where higher degree nodes are connected to lower degree nodes. The best linear fit is overlaid in red.

quadratic fit, a feature commonly present in scale-free networks where larger degree nodes have a disproportionately larger betweenness centrality.[31] Similar to above, outliers from this correlation are labelled in red or blue in figure 8 with the group of nodes in red signifying areas that have higher than expected betweenness centrality and are therefore deemed to be essential to the functioning of the system.

We can see that a similar set of nodes is highlighted by the strength–degree relationship (figure 6) and the betweenness centrality one (figure 8), which is a common occurrence in networks with high reciprocity.[32] These nodes are well connected, central and experience high traffic. Specifically, the examples of radiology (XR), theatres, neurocritical care (NCCU), general medical and neurosurgical wards show that these ward areas have a disproportionately higher betweenness compared with

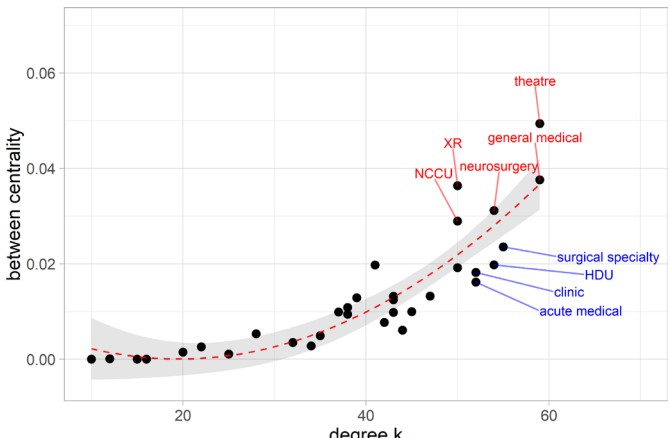

**Figure 8** Betweenness centrality. The distribution of betweenness centrality versus unweighted degree. The relation is fit by a quadratic equation of the degree—as is commonly seen for randomised networks. The ward abbreviations are explained in the online supplementary table.

**Table 1** Hubs and bottlenecks. The hubs and bottlenecks in our system of emergency surgical admissions selected by degree and betweenness centrality

| Hub bottleneck | Hub non-bottleneck | Non-hub bottleneck |
| --- | --- | --- |
| Surgical specialty wards | Acute medical wards | Neurocritical care |
| Neurosurgical wards | High-dependency unit | Radiology |
| General medical wards | Clinics/referral appointments | |
| Theatres | | |

their connectivity. They are therefore more important to the overall structure of the network than their connectivity may suggest. On the other hand, the surgical specialty and acute medical wards have a lower betweenness compared with their connectivity.

We designated the areas with degree in the top 20% of degrees as hubs and the wards with betweenness centrality within the top 20% of betweenness centrality as bottlenecks.[33] This resulted in a set of wards that are both hubs and bottlenecks (hub-bottleneck)—general medical wards, operating theatres, neurosurgical wards—a set that are hubs but not bottlenecks (hub non-bottleneck)—acute medical wards, clinics, specialty surgical wards and high dependency unit (HDU)—and non-hub bottlenecks—radiology and NCCU (see table 1).

### Scale-free and small world networks

Apart from analysing the properties of individual nodes, we used the above calculations to classify our network as a whole. We used the established parameters $\sigma$ and $\omega$ as described by Telesford *et al*[21] and Humphries and Gurney.[20] These two parameters indicate whether a network can be considered to have small-world properties compared with a randomly wired one with similar properties and size. We found that both the original non-categorised network and the categorised network show clear small-world properties. The categorised network had an $\sigma$ of 0.99, where the range of $\sigma$ is from 0 to 1 with values close to 1 representing a small-world network. Also, the value of $\omega$ was 0.0053. Here, the range is from −1 to 1 with the values near 0 representing small-world networks, close to −1 representing lattice networks and values close to 1 are purely random networks. The non-categorised network similarly had $\sigma=0.99$ and $\omega=0.0022$, again reflecting its structure as a small-world network.

However due to the high $\gamma$, representing a more rapid drop of nodes with high degrees than other real-world networks,[12] which tend to have $\gamma$ between 2 and 4, there is a case to be made that this reflects an exponential truncation to a power law.[34] Our system therefore falls into the small-world, scale-free category.

## DISCUSSION

We demonstrated that our system is a complex and densely connected network. This has important consequences as the connections are crucial to behaviour and therefore resilience. Our values for $\sigma$ and $\omega$ as well as the value of $\gamma>3$ (although this is larger than typical for other real-world networks[12]) place our system into the category of small-world networks.[17] These are resilient to shocks, retaining this property even when under attack.[35] However, we also found a dissortative structure; common in technological networks,[36] implying the presence of a node hierarchy where non-hubs are preferentially connected to hubs.[37] Such structures are highly reliant on their hubs[12] making these points of vulnerability. The implication for our system is that the removal of a hub from the network would have a significant impact on the workings of the system and is not easily replaced with existing connections.

It is possible to identify hubs and thus potential bottlenecks of our system (table 1), which is useful in determining where to increase capacity in order to keep the system running smoothly. Combining this information with the areas that have exceptional traffic identified vulnerable areas such as surgical specialty wards, theatres, neurosurgical and general medical wards and radiology. Failure of an area may be due to closure through infectious outbreak, blockage due to high acuity patients or overload of the area. If these events occur in one of these important nodes, the impact on the greater system may be severe and cause problems in unexpected and seemingly unconnected areas. To illustrate, even a small reduction in capacity of radiology, the whole system can experience a significant slowing of the service. Another example is that an infectious outbreak in the general medical wards could lead to significant slowing of the emergency surgical service. With the connection to the elective surgical service via competition for surgical beds such an event may additionally directly influence the ability of the overall surgical service to function.

Many hospitals analyse patient transfers and flow between wards with traditional methods; however, our approach of using network analysis allows us to look at individual areas while taking the whole system into account. The hub bottlenecks are not simply the nodes with the most traffic but are areas essential to the smooth function of the system. The identification of these areas cannot be gained by traditional patient movement counts, but network analysis can provide these additional insights.

The analysis of the traffic between nodes related to connectivity identified areas in the hospital with higher than expected traffic (eg, orthopaedic and surgical specialty wards and NCCU). This technique may help focus future improvement measures to reduce any delays or improve capacity. The fact that in the whole system well-connected wards have disproportionately more traffic is a reflection of the nature of medical services where there is high throughput through certain nodes that are heavily used by different groups and therefore constitute the

base of clinical activity, whereas some subspecialty nodes are only accessed from a small group of nodes for very specific patients.

In the graphical representation figure 4, we observed the expected importance of the particular nodes such as radiology, the theatre complex and surgical wards and particularly neurosurgery (relatively self-contained in our hospital). Unexpectedly, the medical wards, including elderly care also contributed significantly to the functionality of the system. In our setting, this may be due to a variety of situations. These patients either reflect the frequent use of medical areas for surgical overflow, incorrect admission triage, the development of surgical problems during the stay or most likely represent a combination of these. Only the first two possibilities are potential problems for patient care, the third reflects admission to an appropriate location. Investigations into the reasons for the use of medical wards in the management of emergency surgical patients is warranted and will be explored in further work.

In figure 6, nodes with high strength $s >> \bar{w}k$ (where s is the strength, $\bar{w}$ is the average weight and k is the degree) such as theatres or surgical specialty wards represent areas that have high traffic but relatively limited connectivity in contrast to such as IR and rehab which have low traffic compared with their connectivity. This is of interest as it implies that any impediment of traffic through these high traffic areas will have a strong influence on the other areas associated with them. Their influence on the system is therefore more relevant to individual patients' pathways rather than overall behaviour.

Our network representation was developed using retrospective electronic patient records specifically location data which will always contain noise. It is not clear that the overall network structure would be preserved under severe shock conditions (eg, a major incident might severely reduce theatre capacity, an area of vulnerability from our findings). Under such extreme circumstances, the overall configuration of the hospital may transiently be different from what we found under business-as-usual conditions. However, these are rare, and more relevant shocks would be expected to be captured in our model.

Our model is an aggregate of data over the whole time period and does not describe the dynamic behaviour of the system, for example under conditions of hospital strain. It is possible however to construct time-series of networks[9] and it may be that such an approach could demonstrate the actual response of the system to external perturbation. Such an approach might also be a useful framework for forecasting critical bed status.

While this project only considered a single trust and the subsystem of emergency surgical care within it, the project showed the feasibility and potential uses of this type of analysis. We expect that our specific network is unique to our system, commonly seen in settings where network analysis is used, but that the methodology can be transferred to a more general setting including the analysis of regional referral systems.

Our work constitutes a methodology study to assess feasibility of the network approach to understanding the complex system of healthcare provision and hopefully will provide a basis for use in improvement projects. It is beyond the scope of this paper to address specific improvement projects but hopefully this will a be a future use for this type of analysis.

Despite all these limitations, our data appear to show a full picture of the system and give us initial insights in how we can use this approach to inform decisions on service improvement and planning. It would be interesting to compare the configuration of our institution with others so better understand how different hospital configurations might influence potential system resilience.

## CONCLUSIONS

We were able to use electronic healthcare records to create a system representation of our service and demonstrated that emergency surgical services are complex systems with scale-free, dissortative small-world network properties. Such networks are robust overall but may be vulnerable to attack at critical hubs. We were able to identify system bottlenecks, and this may form the basis to inform service improvement initiatives in a more holistic way. This analysis allowed us to show that new insights into the structure and vulnerabilities of a system of care can be gained by combining network analysis and electronic care records. In the future, we hope to use this work to support specific improvement projects and extend the project by considering seasonal effects and use the model to better understand the systems behaviour under strain.

**Acknowledgements** The authors would like to thank Daniel Stubbs for useful discussions and Shaun Hyett for technical assistance in extracting the EHR data. We would also like to thank the Healthcare Design group in the Engineering Department at University of Cambridge for their helpful input.

**Contributors** Both authors (AE and KK) contributed to the conception and design of the work; the acquisition, analysis and interpretation of data; the drafting of this manuscript and give final approval of the version to be published.

**Funding** KK was supported by a NIHR academic clinical fellowship.

**Competing interests** None declared.

**Patient consent for publication** Not required.

**Provenance and peer review** Not commissioned; externally peer reviewed.

**Data availability statement** Data are available upon reasonable request.

**ORCID iDs**
Katharina Kohler http://orcid.org/0000-0003-1919-0193
Ari Ercole http://orcid.org/0000-0001-8350-8093

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
