## [Reviewer comments · BMJ Open]

ARTICLE DETAILS

TITLE (PROVISIONAL)	Can network science reveal structure in a complex healthcare system? A network analysis using data from emergency surgical services.
AUTHORS	Kohler, Katharina; Ercole, Ari

VERSION 1 – REVIEW

REVIEWER	Ara Darzi Imperial College London, UK
REVIEW RETURNED	30-Sep-2019

GENERAL COMMENTS	The authors have conducted a network analysis in their institution for patients undergoing emergency surgical procedures. The aim of which was to identify possible areas for service improvement on a 'grander', more holistic scale. Whilst an increased appreciation of network analysis may add to clinical research and practice, the specific aims of this paper remain unclear, but rather suggest network analysis was applied because this was feasible in this case. As such, the message for clinicians and researchers is not novel; essentially that acute admissions are organisationally complex and require multiple specialties and departments. Issues of nodes, centrality and betweenness were offered but what do they actually mean here and what can we do about them? The nature of the network analysis was very basic and the discussion did not add anything new about emergency surgery. These limitations preclude from appropriateness for publication. A few minor grammatical errors also exist in the manuscript: • Introduction, paragraph 1: 'Admission pressure can only be accommodated if inpatient ow and discharges' - sentence structure• paragraph 3: 'Network science, where the components of a system are represented by nodes and the connections by edges linking them, provides a method to investigate such systems and has been widely applied other disciplines' - sentence structure• 'We attempted to construct a network representation of emergency surgical movements from patient movement data from our comprehensive Electronic Hospital Record (EHR) to give us a fuller understanding of the real architecture of the system of emergency surgical admissions to allow us to identify bottlenecks and vulnerabilities in the service structure' - sentence structure• 'Methods: further ethical approval was needed at no patients were directly contacted' - grammatical error• Results: 'Betweenness centrality' – spelling – spelt betweenness through the manuscript One of the main limitations of this article is the lack of external validity, although the manuscript acknowledges this, the availability of services at their institution will result in differences in patient flow; the availability of neurosurgery and neuro-ICU is a prime example,
---

	also institutions with designated burns units will likely have differing flow hierarchy, etc.... An expansion on this should be made. What is unclear is if this was one data gathered from one trust or one hospital. Discussion, paragraph 4: the authors elaborate on reasons for medical and elderly care contributing as important nodes and state reasons such as surgical overflow, etc.... They should also consider whether medical optimisation prior to surgery could be a reason; e.g. cardiology required for PPM insertion prior to surgery or orthogeriatrics being the prime example; certain institutions also have services for vascular patients with diabetic feet - managed much better on medical wards with surgical intervention sought after appropriate optimisation. Comments on this should be included as differing institutions may admit surgical patients under medical teams deliberately for optimisation prior to surgical takeover. The authors have effectively articulated the complex healthcare system at their institution which can identify potential quality improvement projects (not necessarily in detail); examples should be provided for such improvement projects. If one identifies an issue with a certain patient pathway or the disproportionate traffic for certain nodes, what strategies could be set in place to target these; essentially that is the purpose of such an analysis and without being able to produce specific strategies, it does little justice for their extensive work. As the authors have alluded to already – winter pressure (seasonal effects) would be particularly interesting.
--	---

REVIEWER	Janet Long Australian Institute of Health Innovation Macquarie University
REVIEW RETURNED	25-Oct-2019

GENERAL COMMENTS	Thank you for this opportunity to review your work. This is a very interesting use of social network methodology to take a systems perspective on a hospital service. While resilience of a system – its ability to flex with tidal demands without loss of quality and safety of care – is a difficult trait to define, here they have found an ingenious way to test it by mapping patient flow. The concept of small world networks and the well established evidence of their resilience is a useful way to approach this. Overall, the paper is well written but I have suggested a few ways it could be improved.  1. There needs to be a bit more explanation of social network terms to enable a non-expert in SNA to follow the maths. Acronyms need to be explained. Your brief explanation of social network methods is too brief. 2. “A network is formed of interconnected nodes and the number of connections a node has is called the degree of the node. For directed networks, in-degree and out-degree for a specific node can also be calculated by counting the connections in or out. Connections between the nodes are then represented by their strength or degree of a node weighted by the size of the edges” Please define edges and what you mean by the size of the edges. 3. Line 59 p3 “power law distributions with $p(k) \sim k^{-\gamma}$ can be fitted”– the variables need to be defined. You should not assume all your readers will be SNA experts.
--

	 4. On p5 Can you please spell out what C and L stand for in the equation. 5. Typo on Line 17 page 5 “We found both our non-categorised and categorised networks...” 6. The discussion is interesting but needs a bit more explanation to be useful for readers. The paper is likely to appeal to senior managers who are very unlikely to be experts in SNA. Add some concrete examples and see if you can include some practical implications. Line 12 Page 7: Can you give a hypothetical example along the lines of “To illustrate, an infectious outbreak in ward XX would lead to” 7. Line 17 page 7 Many hospitals have information on patient transfers between wards and units. A short section in the discussion would be useful in explaining what SNA adds to straight counts of patient movements – again making it more accessible and relevant to managers dealing with the hospital’s capacity. 8. Line 31 page 7 Please define the variables in the equation or describe the relationship in words. 9. Line 42 page 7 Word missing: “configuration of the hospital may transiently different from” 10. I really like the suggestion of constructing a time series of networks. It would be quite a big investment in time but would certainly show resilience in action.
--	---

REVIEWER	Joachim P Sturmberg University of Newcastle Australia
REVIEW RETURNED	27-Oct-2019

GENERAL COMMENTS	This is an important paper. The network science approach is highly appropriate for the topic, and the authors are commended for their effort. The paper is describing the network approach clearly though in rather technical terms. I have no troubles with it, and indeed think it is important to do so. However, I also think it is a point of weakness as it leaves many readers unfamiliar with network and complexity science backgrounds stranded. The paper will gain accessibility - and thus impact - by adding "plain English" descriptions to each of the concepts in the results section. I would regard the addition of a Box that provides a clinically oriented story about the point, like: Patients with xxx move to yyy before going to zzz and then home again, whereas patients with AAA move ... or in terms of the bottleneck findings: what creates the bottleneck, how are bottlenecks resolved etc. as a simple and efficient way to provide this clarity.
--

VERSION 1 – AUTHOR RESPONSE

Reviewer: 1

Reviewer Name: Ara Darzi

The authors have conducted a network analysis in their institution for patients undergoing emergency surgical procedures. The aim of which was to identify possible areas for service improvement on a ‘grander’, more holistic scale.

Whilst an increased appreciation of network analysis may add to clinical research and practice, the specific aims of this paper remain unclear, but rather suggest network analysis was applied because this was feasible in this case. As such, the message for clinicians and researchers is not novel; essentially that acute admissions are organisationally complex and require multiple specialties and departments. Issues of nodes, centrality and betweenness were offered but what do they actually mean here and what can we do about them? The nature of the network analysis was very basic and the discussion did not add anything new about emergency surgery. These limitations preclude from appropriateness for publication.

Thank you for your comments. While we understand your concerns about direct clinical applicability and use, in this paper we attempted to use a methodology that has widely been used to characterise and better understand complex systems to characterise a clinical service, which is a novel approach and it is gratifying that the other reviewers are in agreement with us on this point.

There can be little doubt that understanding and improving service provision at a hospital level is important yet there is currently no holistic and principled way of representing such data. We do not claim that our result (in terms of the fact that the system is complex) is novel: we do however claim that our methodology is both novel (as applied to this context) and offers a potential solution to this. We demonstrate for the first time a method which may quantitatively capture this and is therefore appropriate for future evaluation of service and the impact of shocks or attempts at service improvement or reconfiguration, however the practical clinical applications of our project for improvement have not yet been implemented. Applications of our methodology will form the basis of future work and publications- what we present here is methodological and therefore of scientific interest.

The reviewer suggests that our analysis was 'very basic'. We feel that this is not correct- we present network parameters which are the de facto metrics from network science as presented in multiple respected papers in the field. Unfortunately, it is not clear from the comment what more sophisticated mathematical analysis the reviewer had in mind.

We have added additional statements to further explain the aim and state of our project (introduction paragraph 4). In response to the suggestions by BMJ Open editorial staff we have also changed the title, as mentioned above, which hopefully further addresses the reviewers concerns and clarifies the aims of the paper.

Additionally, as the stated scope of BMJ Open includes health services research we feel that this paper is well suited for publication in this journal.

A few minor grammatical errors also exist in the manuscript:

- Introduction, paragraph 1: 'Admission pressure can only be accommodated if inpatient ow and discharges' - sentence structure*
- paragraph 3: 'Network science, where the components of a system are represented by nodes and the connections by edges linking them, provides a method to investigate such systems and has been widely applied other disciplines' - sentence structure*
- 'We attempted to construct a network representation of emergency surgical movements from patient movement data from our comprehensive Electronic Hospital Record (EHR) to give us a fuller understanding of the real architecture of the system of emergency surgical admissions to allow us to identify bottlenecks and vulnerabilities in the service structure' - sentence structure*
- 'Methods: further ethical approval was needed at no patients were directly contacted' - grammatical error*

- *Results: 'Betweenness centrality' – spelling – spelt betweenness through the manuscript*

We have addressed the grammatical and spelling problems in the revision.

One of the main limitations of this article is the lack of external validity, although the manuscript acknowledges this, the availability of services at their institution will result in differences in patient flow; the availability of neurosurgery and neuro-ICU is a prime example, also institutions with designated burns units will likely have differing flow hierarchy, etc.... An expansion on this should be made. What is unclear is if this was one data gathered from one trust or one hospital.

As we state in the manuscript, this was data from 'Cambridge University Hospitals NHS Foundation Trust, a tertiary care hospital' but we have explicitly clarified this to state that the data collection was from a single centre (Methods paragraph 1) and added an additional paragraph discussing the fact that this single-centre study can be seen as a feasibility study with the potential to be applied in both other centres or in regional service studies (Discussion paragraph 9). Quantitative generalisability is not addressed in our paper- indeed in a sense since we propose this methodology as metric of hospital configuration and performance, we expect the results to vary in other centres.

Discussion, paragraph 4: the authors elaborate on reasons for medical and elderly care contributing as important nodes and state reasons such as surgical overflow, etc.... They should also consider whether medical optimisation prior to surgery could be a reason; e.g. cardiology required for PPM insertion prior to surgery or orthogeriatrics being the prime example; certain institutions also have services for vascular patients with diabetic feet - managed much better on medical wards with surgical intervention sought after appropriate optimisation. Comments on this should be included as differing institutions may admit surgical patients under medical teams deliberately for optimisation prior to surgical takeover.

Thank you for the comment on the findings of the analysis. In our clinical experience it is rare in our trust for patients to receive optimisation prior to emergency surgery on a medical ward. In our centre it is also not common practice to deliberately admit surgical patients to medical wards unless their main presenting complaint is medical.

In future work we aim to perform further analysis to investigate the use of medical wards by our emergency surgical patients. We have added additional information on how the medical ward usage may have arisen in our specific setting. (Discussion paragraph 5).

The authors have effectively articulated the complex healthcare system at their institution which can identify potential quality improvement projects (not necessarily in detail); examples should be provided for such improvement projects. If one identifies an issue with a certain patient pathway or the disproportionate traffic for certain nodes, what strategies could be set in place to target these; essentially that is the purpose of such an analysis and without being able to produce specific strategies, it does little justice for their extensive work.

We appreciate this comment as this is exactly the rationale for our work, which we hope will lend itself to informing improvement projects. Evaluation of potential solutions is outside the scope for this work, but could include changes to the elective surgical workload, distributing capacity between specialties more dynamically and changing post-operative pathways. However, since we have not collected data on the specific effects, we would be reluctant to speculate on the effectiveness of such strategies.

As the authors have alluded to already – winter pressure (seasonal effects) would be particularly interesting.

Thank you for this comment, we feel that our project would be very well suited to help understand winter pressures and hopefully be useful in developing strategies to better alleviate the consequences.

Reviewer: 2

Reviewer Name: Janet Long

Thank you for this opportunity to review your work. This is a very interesting use of social network methodology to take a systems perspective on a hospital service. While resilience of a system – its ability to flex with tidal demands without loss of quality and safety of care – is a difficult trait to define, here they have found an ingenious way to test it by mapping patient flow. The concept of small world networks and the well established evidence of their resilience is a useful way to approach this.

Thank you for your kind comments! We have addressed your suggestions/concerns as outlined below:

Overall, the paper is well written but I have suggested a few ways it could be improved.

1. There needs to be a bit more explanation of social network terms to enable a non-expert in SNA to follow the maths. Acronyms need to be explained. Your brief explanation of social network methods is too brief.

In response to this comment, we have added additional explanation to the “Model – Network terminology” section. Additionally, in response to your comment and a comment by reviewer 3 we have added an additional figure describing the process of creating the network model and its analysis.

Indeed our original discussion was significantly longer, however due to word count limitations and literature being freely available (referenced in the section) we feel that further explanation is beyond the scope of this paper. We hope that the additional paragraphs and figure clarify the concepts and process.

2. “A network is formed of interconnected nodes and the number of connections a node has is called the degree of the node. For directed networks, in-degree and out-degree for a specific node can also be calculated by counting the connections in or out. Connections between the nodes are then represented by their strength or degree of a node weighted by the size of the edges” Please define edges and what you mean by the size of the edges.

Thank you for this comment, we have added additional explanation of these concepts, both in the text (Model section paragraph1) and with the additional figure suggested by reviewer 3.

3. Line 59 p3 “power law distributions with $p(k) \sim k^{-\gamma}$ can be fitted”– the variables need to be defined. You should not assume all your readers will be SNA experts.

4. On p5 Can you please spell out what C and L stand for in the equation.

Thank you for pointing this out, this was an oversight on our part. We have added the variable explanations (Model section paragraph 2 & 5).

5. Typo on Line 17 page 5 “We found both our non-categorised and categorised networks...”

This has been corrected.

6. The discussion is interesting but needs a bit more explanation to be useful for readers. The paper is likely to appeal to senior managers who are very unlikely to be experts in SNA. Add

some concrete examples and see if you can include some practical implications. Line 12 Page 7: Can you give a hypothetical example along the lines of "To illustrate, an infectious outbreak in ward XX would lead to"

We have added a few examples of this kind into paragraph 2 of the discussion. Hopefully this paints a clearer and fuller picture.

7. Line 17 page 7 Many hospitals have information on patient transfers between wards and units. A short section in the discussion would be useful in explaining what SNA adds to straight counts of patient movements – again making it more accessible and relevant to managers dealing with the hospital's capacity.

In response to your comment we have added a paragraph in the discussion section (paragraph 3) to better illustrate this point.

8. Line 31 page 7 Please define the variables in the equation or describe the relationship in words.

Again, this was an oversight, thank you for your comment. We have corrected this with the definitions of each variable included.

9. Line 42 page 7 Word missing: "configuration of the hospital may transiently different from"

This has been corrected.

10. I really like the suggestion of constructing a time series of networks. It would be quite a big investment in time but would certainly show resilience in action.

Thank you for your comment. This direction is one of the next steps in our project and we are looking forward to working on it.

Reviewer: 3

Reviewer Name: Joachim P Sturmberg

This is an important paper. The network science approach is highly appropriate for the topic, and the authors are commended for their effort.

The paper is describing the network approach clearly though in rather technical terms. I have no troubles with it, and indeed think it is important to do so. However, I also think it is a point of weakness as it leaves many readers unfamiliar with network and complexity science backgrounds stranded.

The paper will gain accessibility - and thus impact - by adding "plain English" descriptions to each of the concepts in the results section. I would regard the addition of a Box that provides a clinically oriented story about the point, like: Patients with xxx move to yyy before going to zzz and then home again, whereas patients with AAA move ... or in terms of the bottleneck findings: what creates the bottleneck, how are bottlenecks resolved etc. as a simple and efficient way to provide this clarity.

I am looking forward to your revisions.

Thank you for your review and comments. In response to your comment we have added a figure (Figure 1 in the revision) to illustrate the process of creating the network and the analysis of such a network using example patient pathways. We hope that the generic examples, rather than specific examples of network, provide a clearer picture of the method by which the network was created and what the nodes and edges represent. We have also expanded the methodology section in response to your comments and the comments of reviewer 2 (Section Model – network terminology and variables).

VERSION 2 – REVIEW

REVIEWER	Janet Long Australian Institute of Health Innovation, Macquarie University, Australia
REVIEW RETURNED	06-Dec-2019

GENERAL COMMENTS	Thank you for carefully addressing the points I raised. I think the paper is much improved and am recommending it's acceptance.
---

REVIEWER	Joachim P Sturmberg UoN-Australia
REVIEW RETURNED	07-Dec-2019

GENERAL COMMENTS	Thanks for the clarifications in your revision - I will keep an eye on your paper's impact.
---